# Detection of Low-Energy X-rays Using YSO Scintillation Crystal Arrays for GRB Experiments

**Minbin Kim** [1,2], **Jakub Ripa** [3], **Il H. Park** [1,2,*], **Vitaly Bogomolov** [4,5], **Søren Brandt** [6], **Carl Budtz-Jørgensen** [6], **Alberto J. Castro-Tirado** [7,8], **Sheng-Hsiung Chang** [9], **Yenyun Chang** [10], **Chia Ray Chen** [9], **C.-W. Chen** [10], **Pisin Chen** [9,11], **Paul Connell** [12], **Chris Eyles** [12], **Georgii Gaikov** [1], **Gihan Hong** [1], **Jian Jung Huang** [10], **Ming-Huey Alfred Huang** [13], **Soomin Jeong** [14], **Jieun Kim** [15], **Jik Lee** [16], **Heuijin Lim** [17], **Chih-Yang Lin** [9], **Tsung-Che Liu** [18], **Jiwoo Nam** [9,11], **Mikhail Panasyuk** [4,5,†], **Vasily Petrov** [4], **Victor Reglero** [12], **Juana M. Rodrigo** [12], **Sergey Svertilov** [4,5], **Nikolay Vedenkin** [1], **Ming Zu Wang** [10] **and Ivan Yashin** [4]

1   Department of Physics, Sungkyunkwan University, 2066 Seobu-ro, Suwon 16419, Korea;
    happykmb@gmail.com (M.K.); georgy.gaykov@gmail.com (G.G.); hgh987321@naver.com (G.H.);
    vnn.space@gmail.com (N.V.)
2   Institute of Science and Technology in Space, Sungkyunkwan University, 2066 Seobu-ro, Suwon 16419, Korea
3   Department of Theoretical Physics and Astrophysics, Faculty of Science, Masaryk University, 60177 Brno,
    Czech Republic; jakub.ripa@ttk.elte.hu
4   Skobeltsyn Institute of Nuclear Physics of Lomonosov, Moscow State University, Leninskie Gory,
    119234 Moscow, Russia; vit_bogom@nm.ru (V.B.); panasyuk@sinp.msu.ru (M.P.); vas@srd.sinp.msu.ru (V.P.);
    sis@coronas.ru (S.S.); ivn@eas.sinp.msu.ru (I.Y.)
5   Physics Department of Lomonosov Moscow State University, Leninskie Gory, 119234 Moscow, Russia
6   National Space Institute, Technical University of Denmark, 2800 Kgs. Lyngby, Denmark;
    sb@space.dtu.dk (S.B.); carl@space.dtu.dk (C.B.-J.)
7   Instituto de Astrofisica de Andalucia (IAA-CSIC), P.O. Box 03004, 18080 Granada, Spain; ajct@iaa.es
8   Departamento de Ingenieria de Sistemas y Automatica (Unidad Asociada al CSIC), Universidad de Malaga,
    29016 Malaga, Spain
9   National Space Organization, 9 Prosperity 1st Road, Hsinchu Science Park, Hsinchu 30078, Taiwan;
    judean@nspo.narl.org.tw (S.-H.C.); chiaray@nspo.narl.org.tw (C.R.C.); pisinchen@phys.ntu.edu.tw (P.C.);
    sunnylin@nspo.narl.org.tw (C.-Y.L.); jwnam@phys.ntu.edu.tw (J.N.)
10  Department of Physics and Graduate Institute of Astrophysics, National Taiwan University, 1 Roosevelt Road,
    Taipei 10617, Taiwan; gixd@hep1.phys.ntu.edu.tw (Y.C.); cwchen@nspo.narl.org.tw (C.-W.C.);
    njinee@hotmail.com (J.J.H.); mwang@phys.ntu.edu.tw (M.Z.W.)
11  Leung Center for Cosmology and Particle Astrophysics, National Taiwan University, 1 Roosevelt Road,
    Taipei 10617, Taiwan
12  GACE, Edif. de Centros de Investigacion, Universidad de Valencia, Burjassot, 46100 Valencia, Spain;
    Paul.Connell@uv.es (P.C.); cje@star.sr.bham.ac.uk (C.E.); Victor.Reglero@uv.es (V.R.);
    juana.m.rodrigo@hotmail.com (J.M.R.)
13  Department of Energy Engineering, National United University, 2, Lienda Road, Miaoli 36063, Taiwan;
    mahuang@nuu.edu.tw
14  Agency for Defense Development, Yuseong P.O. Box 35, Daejeon 34186, Korea; soominjeong@gmail.com
15  Division of Atmospheric Sciences, Korea Polar Research Institute (KOPRI), 26 Songdomirae-ro, Yeonsu-gu,
    Incheon 21990, Korea; jekim@kopri.re.kr
16  The Center for High Energy Physics, Kyungpook National University, 80 Daehak-ro Buk-gu, Daegu 41566,
    Korea; jiklee999@gmail.com
17  Dongnam Institute of Radiological & Medical Sciences, Jwadong-gil 40, Jangan-eup, Gijang-gu, Busan 46033,
    Korea; heuijin.lim@gmail.com
18  Department of Electrophysics, National Yang Ming Chiao Tung University, 1001, University Road,
    Hsinchu 300, Taiwan; diewanger@gmail.com
*   Correspondence: ilpark@skku.edu
†   Deceased prior to submission of this paper.

**Abstract:** We developed an X-ray detector using 36 arrays, each consisting of a 64-pixellated yttrium oxyorthosilicate (YSO) scintillation crystal and a 64-channel multi-anode photomultiplier tube. The X-ray detector was designed to detect X-rays with energies lower than 10 keV, primarily with the aim of localizing gamma-ray bursts (GRBs). YSO crystals have no intrinsic background, which is advantageous for increasing low-energy sensitivity. The fabricated detector was integrated into UBAT, the payload of the Ultra-Fast Flash Observatory (UFFO)/*Lomonosov* for GRB observation.

The UFFO was successfully operated in space in a low-Earth orbit. In this paper, we present the responses of the X-ray detector of the UBAT engineering model identical to the flight model, using $^{241}$Am and $^{55}$Fe radioactive sources and an Amptek X-ray tube. We found that the X-ray detector can measure energies lower than 5 keV. As such, we expect YSO crystals to be good candidates for the X-ray detector materials for future GRB missions.

**Keywords:** gamma-ray burst; YSO; UBAT; UFFO

## 1. Introduction

Gamma-ray bursts (GRBs) are known to be the most luminous explosions and could serve as signposts throughout the distant universe right up to the post-dark age. GRBs emit not only high-energy photons but also span nine orders of magnitude in photon energy, which provides a prime opportunity for synoptic observations. Owing to their association with gravitational waves and possibly with ultra-high-energy particles, GRBs are also believed to be powerful tools for multi-messenger astrophysics [1,2].

GRBs have been detected at a redshift of z > 6 [3,4]. Perspective research on the most luminous sources in the sky is expected to explore the early universe of redshifts beyond z = 6 [5], whereas supernovae SN 1a have been observed up to z = 2.26 [6].

As a leading observatory for the discovery of high-energy transients since 2004 [7], *Swift* found the first GRBs at z = 6, 7, 8, and 9. The *Swift* observatory pinpointed the locations of short-GRB afterglows [8] and identified nearby GRBs with and without supernovae [9]. The Fermi gamma-ray telescope [10], launched in 2008 with an unrivaled spectral range, provides new data on gamma-ray emissions from GRBs. Other space missions for the observation of GRBs are INTEGRAL [11], AGILE [12], Suzaku [13], MAXI/ISS [14], the Interplanetary Network (IPN) [15,16] satellites, and the recent but short-lived BDRG/*Lomonosov* [17] and UFFO/*Lomonosov* [18].

If forthcoming missions are to realize a deeper understanding of GRBs, there is a need not only for large number of GRBs, but also for those with high z-values. For example, the detection of ~1000 GRBs/year is necessary for selecting 50 and 5 GRBs at z > 10 and 20, respectively. Thus, a mission would have to have a lifetime of at least five years if it were to provide answers to questions related to the early universe [19,20]. Future experiments are expected to increase the detection area and/or volume of X-rays and the aperture of UV/optical/IR telescopes, as well as improve the photometric, temporal, and spectral sensitivities.

There has been an attempt to improve temporal and spectral sensitivities through the use of a fast-slewing optical telescope and a low-energy X-ray-sensitive telescope. The Slewing Mirror Telescope (SMT) aboard the *Lomonosov* spacecraft operated successfully in space [18,21] and required approximately <2 s to target a GRB source for follow-up observations following a trigger from X-rays [22]. In the following, we briefly describe the X-ray detector on the *Lomonosov* spacecraft and present the dynamic range of its energy response.

## 2. UFFO/*Lomonosov* for Observation of GRBs

The Ultra-Fast Flash Observatory (UFFO) is a new concept in space telescopes for measuring the early photons from a GRB down to sub-second timescales. After detecting X/gamma rays from a GRB, the main idea is to quickly target it with a UV/optical telescope through a slewing mirror. This was proposed in 2009 [23]. The first mission of the UFFO project UFFO/*Lomonosov* [18,21] was developed and fabricated onboard the *Lomonosov* satellite, which was launched on 28 April 2016 [24]. After the launch, calibration of the satellite and payloads, including the UFFO/*Lomonosov*, was carried out for approximately seven months, but unfortunately, the operation was ended before any GRB data could be acquired because of a power supply failure in the satellite.

UFFO/*Lomonosov* consists of two telescopes: a UFFO Burst Alert and Trigger telescope (UBAT) [25] and an SMT [26,27]. The UBAT is an X-ray trigger telescope, and the SMT is a UV/optical telescope, as shown in Figure 1. The UBAT has a wide field of view (FOV) of $90° \times 90°$. It is responsible for the localization of GRBs through the detection of X-rays and the provision of a trigger to the SMT for rapid follow-up optical/UV observation. The SMT only moves the slewing mirror once it has been triggered by the UBAT, and can start the UV/optical observation within 2 s.

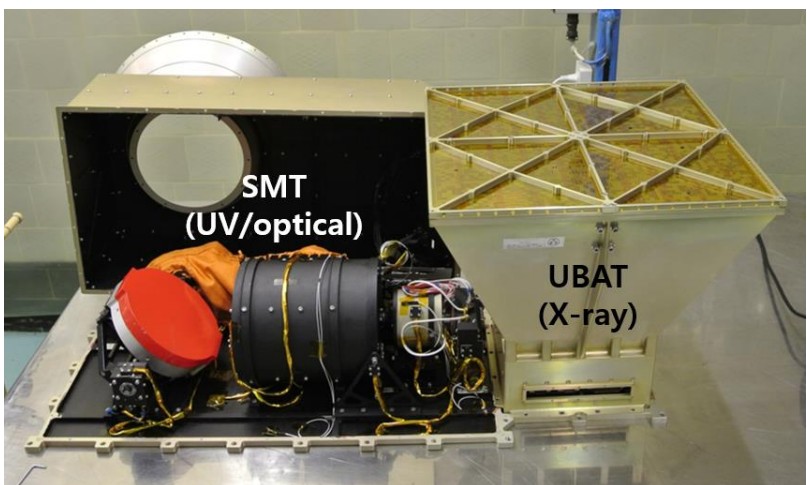

**Figure 1.** The UFFO/*Lomonosov* flight model. It consists of an SMT and a UBAT. The housing, slewing mirror (red cap), telescope (black cylinder), and electronics of the SMT are shown.

As shown at the top of the UBAT in Figure 1, the UBAT is equipped with a coded mask covering the X-ray sky with a wide FOV of 1.8 sr. Its localization accuracy is $\pm 5$ arcmin at more than 7σ. The specifications of the UBAT are summarized in Table 1. Details are described in previous report [25].

**Table 1.** Specifications of the UFFO Burst Alert and Trigger telescope (UBAT).

| UBAT Parameter | Value |
|---|---|
| Aperture | coded mask, random pattern, about 45% open |
| Field of view | 1.8 sr (partially coded) |
| Pointing accuracy | 10 arcmin for $\geq$7σ |
| Detector element | 64-pixellated YSO crystal + MAPMT |
| Number of pixels | 2304 pixels (48 × 48) |
| Detector area (effective area) | 191.1 cm$^2$ (165.5 cm$^2$) |
| Detector thickness | 3 mm |
| Detector energy range | <5–200 keV |
| Detection efficiency | 95.90% at 30–50 keV |

## 3. X-ray Detector of UBAT

### 3.1. YSO Scntillation Crystals for X-ray Detection

The UBAT detector consists of 36 cesium-doped yttrium oxyorthosilicate (YSO, $Y_2SiO_5$:0.04% Ce) scintillation crystal arrays. Each array is pixelated by 64 channels and read out by a multi-anode photomultiplier tube (MAPMT) and associated electronics. There is a total of 2304 pixels or channels. We chose to use a YSO crystal for the conversion of X-rays to UVs because it offers many advantages that make it suitable for our experiments. First, there is no intrinsic background. This increases the possibility of detecting

low-energy X-rays. Next, it has a high light yield of 9.2–10 photons/keV, suitable for UBAT, which has to localize a GRB with only a small number of photons. It also has a fast decay time of ~35 ns [28], which is appropriate for the readout electronics of our analog–digital conversion. In addition, it has good radiation-hard properties [29,30], allowing for at least three years of space operation, as well as mechanical robustness, such as no cleavage planes and non-hygroscopicity.

The size of one crystal pixel is $2.68 \times 2.68 \times 3$ mm$^3$. Five sides of the crystal (all except the one that meets the bottom) were covered with a 200 μm dielectric reflector. The reflector on the sides adjacent to other pixels reduces crosstalk by preventing the movement of photons to neighboring pixels, and the reflector on the top side prevents photon loss by total internal reflection. The total detection area of the UBAT is 191.1 cm$^2$, whereas the effective area is 165.5 cm$^2$ which corresponds to the total area of the YSO crystals. Figure 2 shows the integrated flight model of the UBAT detector.

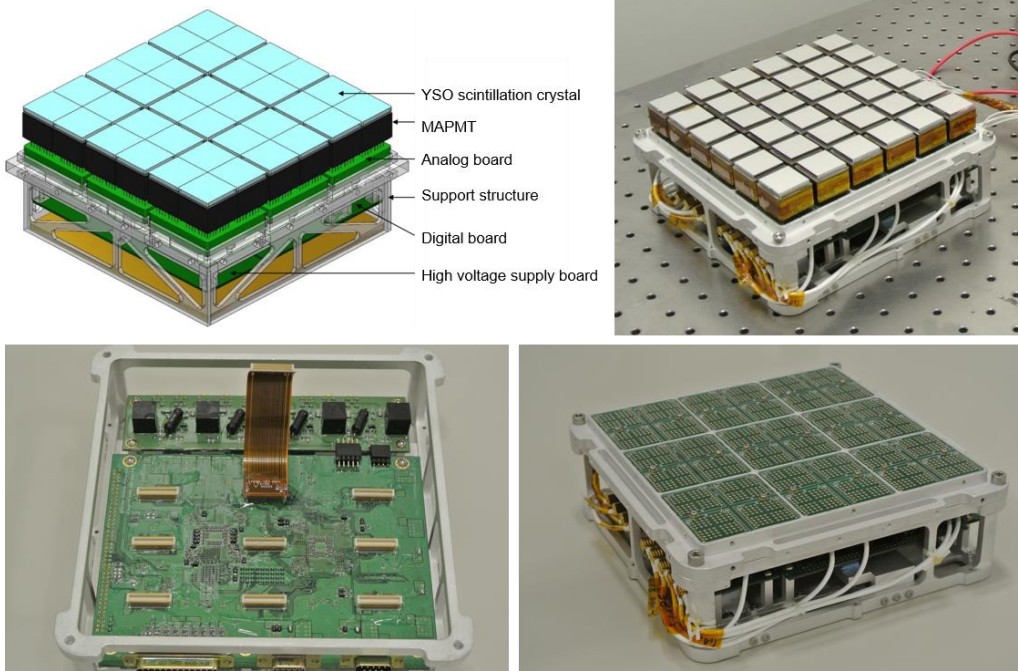

**Figure 2.** UBAT detector. A rendering of an integrated UBAT detector (**top left**) and the integrated flight model of a UBAT detector (**top right**). Bottom part of a UBAT detector. A high-voltage board and a digital board are assembled in the detector structure. One FPCB is connected to the digital board (**bottom left**). Integrated UBAT detector top and bottom part. The nine analog boards are connected to the digital board and the high-voltage supply board, respectively (**bottom right**).

### 3.2. MAPMTs for Electric Signals

UV scintillation lights produced in a YSO crystal array enter a MAPMT (R11265-03-M64) that was developed by RIKEN in collaboration with Hamamatsu Photonics K.K for the JEM-EUSO space experiment [31]. The MAPMT also has $8 \times 8$ pixels, with each pixel size $2.88 \times 2.88$ mm$^2$. There are no gaps between the pixels. Each MAPMT can be readily assembled underneath a YSO crystal array, affixed with a space-qualified optical glue. The window material of the MAPMT is UV-enhanced glass with a thickness of 0.8 mm and a spectral response range of 185–650 nm. Light passing through the window is converted to photoelectrons in an ultra-bialkali photocathode with a spectral response range of 300–650 nm and a maximum quantum efficiency of 43% at a wavelength of 400 nm [32].

*3.3. Electronics*

The UBAT has analog, digital, and high-voltage supply electronics. Each uses space-qualified components and passed all space environment tests. The UBAT includes nine analog boards. One analog board is connected to four MAPMTs and contains four Spatial Photomultiplier Array Counting and Integrating Chips (SPACIROCs), an Application-Specific Integrated Circuit (ASIC) [33], and two high-voltage dividers. One ASIC was paired with one MAPMT with 64 channels. The analog boards perform photon counting and energy measurements through a charge-to-time conversion. Each ASIC acquires data every 2.5 μs, and the double pulse resolution is 50 ns. It also sets parameter values reaching 900 bits, including photon counting and energy measurement thresholds, gain, and channel masking. The thresholds for photon counting and energy measurement can be changed for each MAPMT, whereas the gain adjustment and channel masking are pixel by pixel. These values were pre-programmed but can also be modified by commands sent from the ground.

The UBAT has one digital board, and the main components are two field programmable gate arrays (FPGAs), a static random-access memory (SRAM), and a NOR flash memory. It receives analog signals from analog electronics, converts them into digital values via 8-bit ADCs, and then performs data processing on one FPGA and trigger calculations on the other. Energy is measured by summing eight adjacent channels each with 8-bit data, whereas so-called rate trigger calculations use the total number of photons and therefore only require 1-bit data for each channel. In the case of a positive rate trigger, the imaging of the coded mask pattern starts immediately. Raw data are stored in the SRAM. The NOR flash memory holds the ASIC parameter setting values. In addition, the digital board interfaces with the data acquisition system and the power distribution system of the UFFO/*Lomonosov*. It also monitors the housekeeping parameters, including the voltage, current, and temperature of the UBAT components.

The high-voltage electronic supplies the approximately 1000 V required to drive the MAPMTs. It employs nine DC–DC converters to provide an output voltage of 500–1200 V with a gain of $10^2$. The nine high output voltages are routed to nine analog boards each with four MAPMTs. The default input voltage of the MAPMT is approximately 1000 V and can be adjusted in 100 V increments by sending a command.

## 4. Measurement of Energies with YSO and MAPMT

In addition to the flight model (FM) described above, we have an identical set of UBATs on the ground called the engineering model (EM). When the UBAT FM acquires data in space, it is stored in the SRAM of the digital board and transferred to the satellite through a bus interface. For the report below on the energy response to UBAT, we used the UBAT EM and a standalone data acquisition system that saved data stored in SRAM directly to a laptop using a National Instruments (NI) device.

Figure 3 shows the measurement of energy using an Amptek X-ray tube source [25]. The exposure time for each dataset was 200 s. We identified energy resolutions of $109.80 \pm 3.38\%$, $73.59 \pm 5.34\%$, and $53.05 \pm 2.37\%$ at energies of 8.7, 13.4, and 26 keV, respectively. The energy resolution is defined as $\frac{\Delta E}{E}[\%] = \frac{FWHM}{E} \times 100$. Note that X-ray tubes produce a wide range of energies, including the continuous spectrum of bremsstrahlung X-rays and characteristic lines of X-rays emanating from the target. These scatters from the target degrade the energy resolution of the detector.

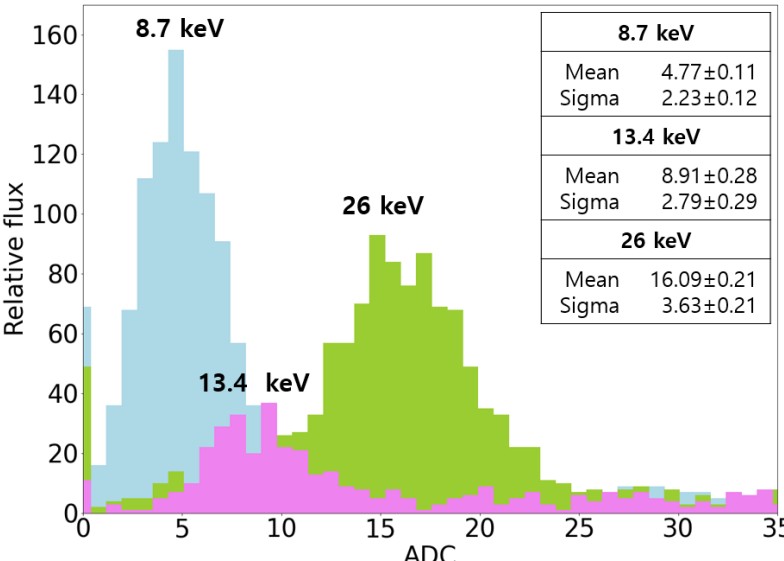

**Figure 3.** Energy measurements of a UBAT in terms of ADC counts using the Amptek X-ray tube with an Au target. Voltages applied to the tube were 10, 15, and 30 kV for energies of 8.7, 13.4, and 26 keV, respectively. In addition, for energies of 13.4 keV and 26 keV, an Mo filter with a thickness of 0.025 mm and a Cu filter with a thickness of 0.3 mm were used, respectively.

Figure 4 shows the measurement setup in which a radioactive source (yellow circle) with a collimator was placed on top of a YSO crystal (silver). The green board is an analog board. The collimator is made of lead and has a hole approximately 1 mm in diameter, which covers an entire YSO crystal module such that the X-rays only enter the pixel that measures the energy. We used an $^{241}$Am source with peaks at 18 and 60 keV, and an $^{55}$Fe source with a peak at 5.9 keV.

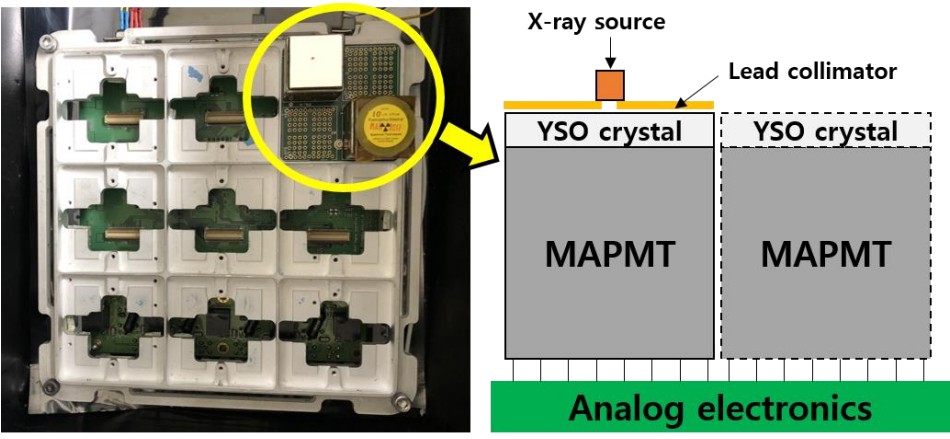

**Figure 4.** Setup for X-ray energy measurement. The top-right corner of the left-hand figure shows a radioactive source (circle) placed on top of one of two YSO scintillation crystals (silver), the MAPMTs, and an analog board below. The right-hand figure is a schematic side-view drawing of the setup with a lead collimator between the X-ray source and the YSO scintillation crystal.

Figures 5 and 6 show the measured energy spectrum in the pixel for the $^{241}$Am and $^{55}$Fe X-ray sources, respectively. The exposure time for one frame was 1 ms, and each data point was acquired for 70 s. Using a conversion factor of 1.837 keV per ADC count, we obtained energy resolutions of 87.59 $\pm$ 12.67%, 85.32 $\pm$ 3.42%, and 29.38 $\pm$ 2.49% at 5.9, 18, and 60 keV, respectively. The linearity of the ADC measurement with respect to the incident energy is plotted in Figure 7 as a function of the energy. The function $\frac{\Delta E}{E}$ can be

expressed as a power law $\left( a\, E^{-b} \right)$, and the values of $a$ and $b$ were found to be 287.95 and 0.519, respectively, which are consistent with measurements at high energies [34].

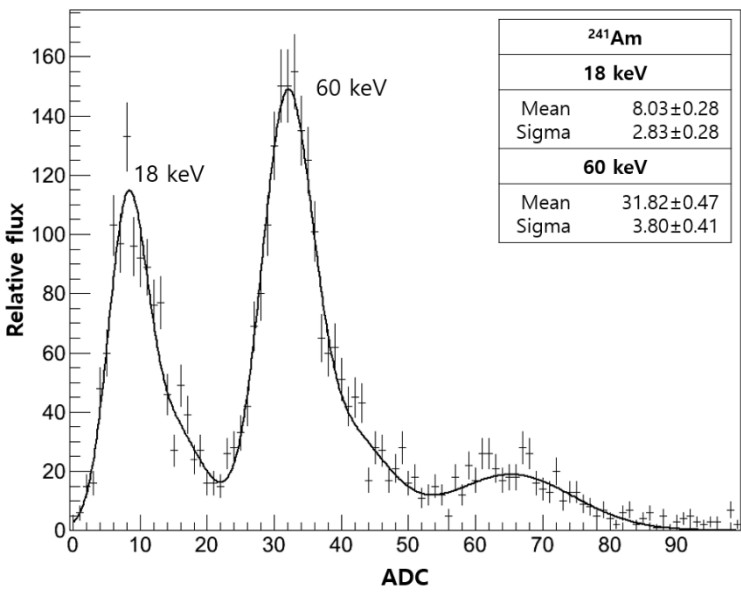

**Figure 5.** Energy measurements with $^{241}$Am X-ray source. ADC counts versus relative flux. $^{241}$Am has two peaks at 18 and 60 keV. The solid line represents a fit to the function of the linear sum of five Gaussians.

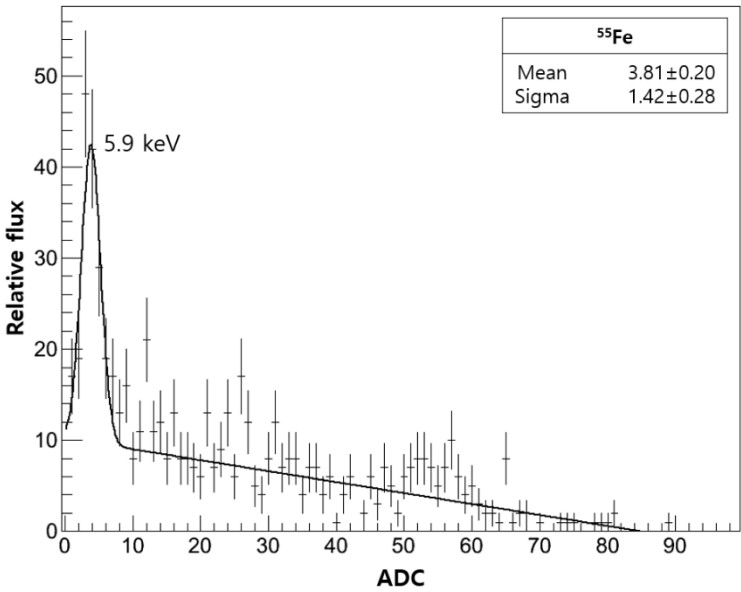

**Figure 6.** Energy measurements with $^{55}$Fe X-ray source. $^{55}$Fe has a peak at 5.9 keV. The solid line represents the fit of the Gaussian plus the straight line.

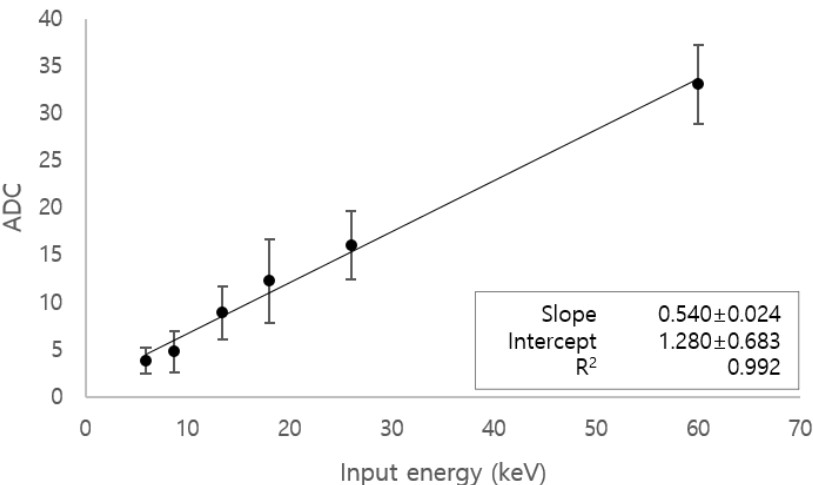

**Figure 7.** Linearity between energy and ADC counts. Conversion factor is 1.837 keV per ADC count.

This result shows that the YSO crystal provided a dynamic range of X-ray energies down to a few keV. The thickness of the crystal used in the UBAT was 3 mm, which allowed us to measure up to 200 keV with an efficiency of >70% [35]. By increasing the thickness of the crystal, the upper bound of the energy measurement could be readily extended to MeV. Therefore, the YSO crystal is a promising detector material that could be utilized in future missions for the observation of GRBs.

## 5. Detection Threshold for the Observation of GRBs

The aim of the UBAT is the localization of GRBs, which depends not only on the flux of the GRBs, but also on the background, which consists primarily of the cosmic X-ray background (CXB). The typical GRB spectrum is well described by the Band function [36] or by the Comptonized model (CPL, exponential cutoff power law) [37]. We adopted the CPL model for a typical long GRB for the fluence spectrum, that is, the time-integrated spectrum over the entire duration of the burst. Such a spectrum was derived using the Fermi GBM Burst Catalog (FERMIGBRST) [38] in [39]. For the differential photon flux $N_E$ in photons cm$^{-2}$ s$^{-1}$ keV$^{-1}$ it holds:

$$N_E = A \left( \frac{E}{E_{piv}} \right)^{\alpha} exp \left[ -\frac{(\alpha + 2)E}{E_{peak}} \right] \tag{1}$$

where $E_{peak}$ is the peak energy; $A$ is a normalization constant; $E_{piv}$ is the pivot energy, which is typically maintained at 100 keV; $\alpha$ is the spectral index; and $E$ is in keV. The values for the typical long GRB are $A = 8.9 \times 10^{-3}$ photons cm$^{-2}$ s$^{-1}$ keV$^{-1}$, $E_{peak} = 183$ keV, and $\alpha = -0.96$ [39]. Figure 8 shows the differential photon flux for a typical long GRB, where the function $N_E$ is multiplied by a factor of 0.5, which approximately corresponds to the fraction of photons blocked by the UBAT's coded mask.

The cosmic X-ray background is well described by two power laws in the range of 2 keV–2 MeV [40]. For an overview of the different background components in low-Earth orbit, see [39]. Figure 8 shows the CXB spectrum multiplied by the FOV of the UBAT, which was 1.85 sr, and by a factor of 0.5, taking into account the blockage by the coded mask. At low energies, the CXB differential photon flux asymptotically approached $N_{CXB} \sim E^{-1.32}$.

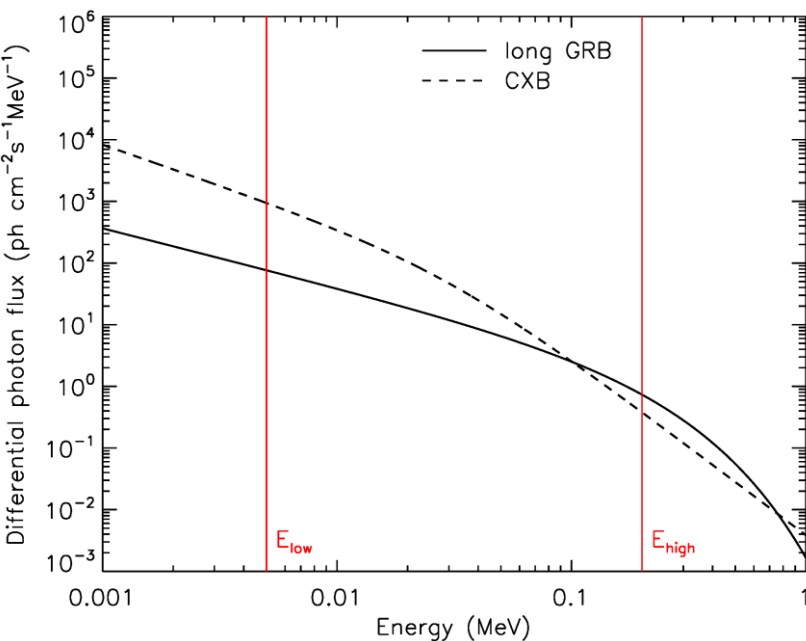

**Figure 8.** Differential photon flux of a typical long GRB (fluence spectrum) and CXB as a function of X-ray energy. Energy limits $E_{low}$ and $E_{high}$, which are 5 and 200 keV here, respectively, define the dynamic range of the detector. The fluxes are multiplied by a factor of 0.5, which approximately corresponds to the fraction of photons blocked by the UBAT's coded mask. The CXB flux is also multiplied by the detector's FOV of 1.85 sr.

Figure 9 shows how the detected integrated photon rates from the GRB and CXB changed with the low-energy threshold of the detector, $E_{low}$, for a fixed $E_{high}$ value of 200 keV. For an energy range of 5–200 keV, the photon detection rate expected from UBAT would be 1.2 photons $cm^{-2} s^{-1}$ from a GRB and 6.4 photons $cm^{-2} s^{-1}$ from the cosmic X-ray background. This estimate of the background is in a good agreement with the measured rate in space from the UBAT, which was 4.5 photons $cm^{-2} s^{-1}$. This smaller count rate may have been due to the absorption of low-energy X-rays by the Kapton tape on top of the coded mask and the black Tedlar that wrapped the crystal to block UV/optical light.

Lowering the detector's energy threshold does not provide a significant gain in the GRB detection, because the GRB spectrum has a power-law slope of $-0.96$ but CXB has a power-law slope of $-1.32$. Thus, the slope of signal-to-noise ratio (SNR) will have a power of $-0.36$. Note that SNR $= \sqrt{A_{det} t_c} \frac{N_{GRB}}{\sqrt{N_{CXB}}}$ where $A_{det}$ is the detector area and $t_c$ is the photon collection time. This shows that a larger detector area offers the advantage of a shorter collection time. A sizable increase in the SNR provides so-called rate trigger followed by image processing of the mask pattern for the localization of GRBs, as implemented in the UBAT trigger algorithm. In Figure 9, we show the SNR of the UBAT for the rate trigger for a detector area of 165.5 $cm^2$ and assuming a photon collection time of 1 s. As shown in the figure, the UBAT rate trigger increased with a lower energy threshold $E_{low}$. Here, we assumed that the prompt GRB spectrum could be extrapolated to a few keV.

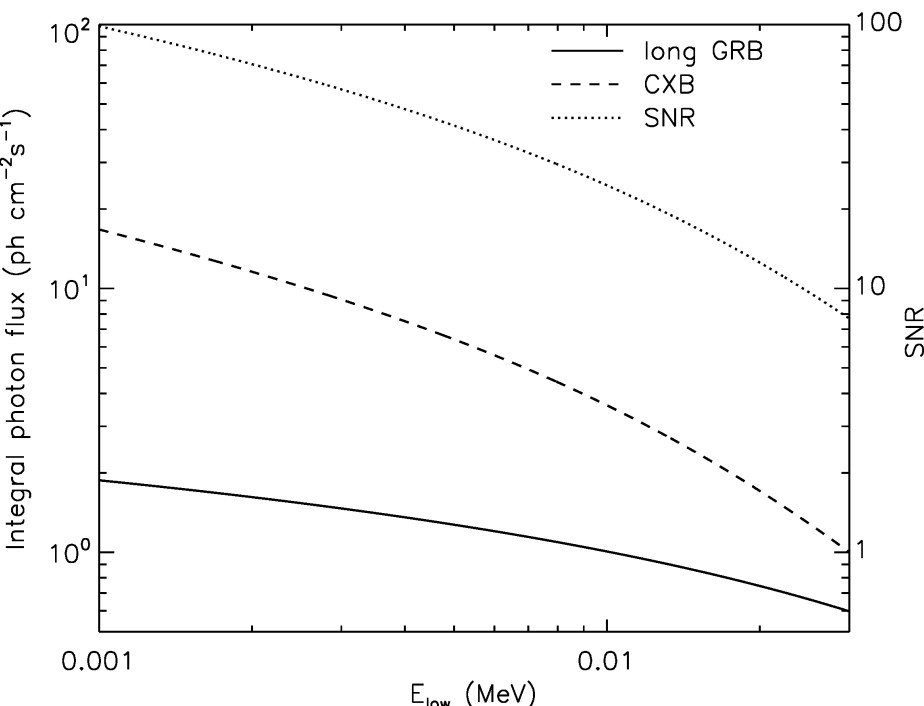

**Figure 9.** Integral photon flux of typical long GRB (fluence spectrum) and CXB as a function of the low-energy threshold, with $E_{high}$ fixed to 200 keV. The right-hand y-axis shows the SNR value of the UBAT assuming a photon collection time of 1 s.

The energy spectrum of a GRB at a higher redshift was shifted towards lower energies, as defined by the formula of $E_{z=z'} = E_z \frac{1+z}{1+z'}$ . The flux decreased as the object moved further away. Although it is a simple assumption, we only considered the shift in energy due to the cosmological redshift. In practice, GRBs exhibit a wide range of luminosities, so there may be high redshift GRBs with a high luminosity but similar intrinsic energy spectra so that the observed fluxes are the same. From this point of view, to investigate the full energy spectrum of GRBs at a high redshift, the dynamic range of the energy measurements should be as wide as possible, especially for low energy thresholds.

## 6. Conclusions

The UBAT, a coded mask-based X-ray telescope carried by the UFFO/*Lomonosov*, employs a YSO scintillation crystal for the detection of X-rays. It was applied for the first time to space experiments and was shown to operate successfully in space. We found from the results of energy measurements on the engineering model of UBAT that a YSO crystal can detect energies of <5 keV. It provides a wide dynamic range in energy measurements from a few keV to MeV using a single detector material. This wide dynamic range is advantageous for GRB observations, especially for energy spectral studies. It also helps in the localization of high-redshift GRBs, as lowering the low energy threshold improves the probability of observing GRBs to a power of $-0.36$ if the GRB's dimming is not crucial.

**Author Contributions:** Test of UBAT, M.K., S.J., J.L. and J.R.; UBAT electronics, M.K., G.G., G.H., J.K. and H.L.; UBAT-coded mask, P.C. (Paul Connell), C.E., V.R. and J.M.R.; theoretical calculations, S.B., C.B.-J. and A.J.C.-T.; PI of UFFO/*Lomonosov*, I.H.P.; PI of *Lomonosov*, M.P.; UFFO/*Lomonosov* space environment test, S.-H.C., Y.C., C.R.C., C.-W.C., P.C. (Pisin Chen), J.J.H., M.-H.A.H., C.-Y.L., T.-C.L., J.N. and M.Z.W.; interface between UFFO and *Lomonosov*, V.B., V.P., S.S., N.V. and I.Y.; writing I.H.P., M.K. and J.R. All authors have read and agreed to the published version of the manuscript.

**Funding:** This study was supported by grants from the Korean National Research Foundation (NRF-2017K1A4A3015188, NRF-2021R1A2B5B03002645, and NRF-2019H1D3A2A02060090). The Russian work was partially supported by ROSCOSMOS grants and by RFFI grant No. 13-02-12175 and

No.15-35-21038 and support from the Development Program of Lomonosov Moscow State University is acknowledged. AJCT acknowledges support from the Spanish MINECO Projects AYA 2009-14000-C03-01/ESP and AYA 2015-71718R (including EU/FEDER funds). The authors thank Taiwan's National Science Council Vanguard Program (100-2119-M-002-025) and Ministry of Science and Technology (MOST) for the funding (104-2811-M-002-160). MBK acknowledges the support from NRF-2015-GPF.

**Informed Consent Statement:** Not applicable.

**Acknowledgments:** In the memory of Mikhail Panasyuk, we deeply appreciate his leadership and contribution to many space projects.

**Conflicts of Interest:** The authors declare no conflict of interest.

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
