# Peer review of "Detection of Low-Energy X-rays Using YSO Scintillation Crystal Arrays for GRB Experiments"

_universe, doi:10.3390/universe7110396_

Round 1
Reviewer 1 Report
Referee report
The article titled "Detection of low energy X-rays using YSO scintillation crystal arrays for high redshift GRB experiments" written by Kim et al. is demonstrating the detection of a few keV X-rays using YSO crystal and a multi anode MPT designed for the UFFO/Lomonosov mission.
Although the article contains detailed and important information about the hardware, I found some results are missing to understand the performance of the instrument in the current draft. I also found that it is a bit overstated about the possibility of detecting high redshift GRBs using a proposing instrument.
Major comments
- How many bits are in the ADC? Please specify in section 2.3.
- In section 2.3 line 187, it is not clear whether the energy threshold can be changed by every single-pixel or not. Please specify in section 2.3.
- Do you have the data showing the gain variation among the MAPMT pixels? What was the standard high voltage setup during the flight operation of UBAT?
- What is the energy resolution of 8.6 keV, 15.9 keV, and 28.8 keV lines in Figure 3?
- What is the energy resolution of 8.6 keV, 15.9 keV, and 28.8 keV lines in Figure 3?
- The title of section 4 is "Energy sensitivity of UBAT." I found it a bit misleading in this title. In this section, the authors are describing the dynamic range of UBAT. It is less clear which part of the description corresponds to the "sensitivity" of UBAT. I would suggest changing the title of the section to "Dynamic range of UBAT" or something equivalent.
- Do you have any spectral data confirming the detection of photons > 100 keV by UBAT?
- Do you have the data showing the linearity between the photon energy and ADC?
- Not only the dynamic range (sensitive to a few keV X-rays) but also the sensitivity of the instrument is an important factor. For example, BeppoSAX WFC and HETE-2 WXM were both sensitive to a few keV X-rays (the energy range was 2-30 keV). However, we knew that those GRB instruments never detected z>10 GRBs. I found the statement "the YSO crystal demonstrates a promising detector material utilized in the future missions for the observation of high-redshift GRBs (line 241-242) and "Because the UBAT detector measures X-rays of energy about 3 times lower than Swift, it can observe higher redshift GRBs, theoretically about 3 times farther away (line 260-262)" are overstated unless the author shows the sensitivity of the UBAT instrument (for example, comparing the sensitivity of the soft events between UBAT and BeppoSAX WFC or HETE-2 WXM).
Minor comments
- line 65: ".. multi-messenger astrophysics". A reference is needed.
- line 68-69: ".. supernovae SN1a cannot reach beyond z=2-3." A reference is needed.
- line 70: ".. since 2004. Swift found .." should be ".. since 2004, Swift found.."
- line 71: ".. the location of short- GRB afterglows," A reference is needed.
- line 72: ".. identified nearby GRBs with and without supernovae." A reference is needed.
- line 208 ".. using an NI device" "NI" should be spelled out (e.g., National Instruments).
- line 233-235: ADC count has the error. What is the confidence level of the error? Please state.
- line 257: "using a 55Fe" of "55" should be upper case.
Reviewer 2 Report
The manuscript entitled "Detection of low energy X-rays using YSO scintillation crystal arrays for high redshift GRB experiments" presents a laboratory evaluation of the UBAT X-ray detector flown on the UFFO/Lomonosov mission. The evaluation is performed on a lab spare of the hardware, as the original mission was not able to detect GRBs due to technical issues with the power supply.
As the global mission combined two quite innovative ideas -- slewing mirror optical telescopes for fast acquisition of transients and very low background YSO scintillators for low-energy X-ray detection -- it is clearly scientifically interesting to also see a detailed account of the detector characterisation. The authors present a laboratory investigation of the detector elements using both an X-ray tube and radioactive samples to characterise the response. The experiment is sound, and the results have scientific merit.
However, in the present state of the paper there are several major points that should be addressed to increase the readability of the manuscript and coherence and clarity of the presentation:
1.: The manuscript is in need of extensive language editing mainly pertaining to English grammar. Obviously this is not a core issue regarding the scientific content, but at the moment the language errors impede readability of the entire manuscript. I give a few examples from the abstract here, the errors are all quite similar (very often missing / wrong articles and / or pronouns), and unfortunately much too numerous to list them all in the report:
Line 46: "We have developed the X-ray detector using 8 × 8 Yttrium Oxyorthosilicate (YSO)..." --> should either be "the UBAT detector" or "a X-ray detector".
Line 49/50: "...thus it has a great advantage to increase low energy sensitivity." --> should be "provides a great advantage for increasing the low energy..."
Line 53: "...and the Amptek X-ray tube..." --> "and an Amptek..."
...as said, many more and similar instances in the entire manuscript.
2.: In many instances, this includes the introduction, but also other parts, the presentation of the results is not at all clear. Most striking examples:
Line 53/54: "We found the sensitivity of the X-ray detector far lower than 5 keV" --> sensitivity measured by which criterion? Do you maybe mean energy threshold? If yes, by which statistical measure?
Line 60ff: "GRB releases in less than a minute the entire energy of the Sun radiating for its lifetime and occur several times every day" --> This is a sentence surely adequate for a popular presentation, and not wrong per se, but in a scientific paper it should really be qualified. All the more as the fluence of GRBs is very directly related to the topic of the manuscript (detection prospects of high-z GRBs with novel scintillators). Please clarify if isotropic equivalent luminosities are what you describe here, and also quantify the total energy budget, including a citation.
Line 62: "GRBs emit not only the highest energy photons" --> This is actually not correct. Though GRBs copiously emit gamma-rays, the highest energy gamma-rays at the moment would probably be those seen by HAWC and LHASSO from potential galactic PeVatrons, tentatively associated with star forming regions / SN sites.
Line 64: For the association of GRBs and GW detections, GW170817 would obviously be the event the authors refer to. Please state that, and provide a citation of the LIGO/VIRGO/Multi-Wavelength paper at least.
Line 66: Although GRBs are observed at high redshifts, the most distant discrete objects presently observed are actually galaxies (see Oesch et al., The Astrophysical Journal. 819 (2): 129.).
Line 68: If the authors really think SN Ia will not be detectable beyond z=2.3, a quote would ne needed. Alternatively, why not rephrase to something like "We thus consider GRBs powerful alternative cosmological probes even if compared to type Ia supernovae."?
Line 70ff: Citations needed for the achievements of the space missions listed.
Line 80: Answering which early Universe questions precisely?
3.: The authors should expand on what precisely is the consequence of the result of their investigation regarding a future mission with this scintillator. How will a potentially lowered threshold (not "sensitivity" as often used in the manuscript) of 5keV translate into an increased detection rate of high redshift bursts? The authors state "about 3 times farther" (Line 261/262), but how will that really hold up? Surely we are not simply talking about redshift here, as we just do not know with certainty if there are any GRBs at z = 25, and how their spectra would look like. So, how will the scaling be at redshifts we have some detections from? This surely can be quantified, e.g. by comparing the spectra of the bursts with the response of the detector.
Reviewer 3 Report
This manuscript reports design of the X-ray detector for energies less than 10 keV on board of UFFO/Lomonosov Gamma-Ray Observatory. My comments below concern scientific rationale for detection of X-ray emission of Gamma Ray Bursts (GRBs) by this and future instruments using similar design.
Line 68: Authors claim that GRBs can be observed up to redshift 30-60 in X-ray. This estimation is highly unrealistic and must be revised. Luminosity of a GRB at z ~ 30 would be ~10^4 times fainter than one at redshift ~1. Even for E_iso ~ 10^54 i.e. comparable to the intrinsically brightest GRBs such as GRB 080916C, GRB 080607, and GRB130505A the observed fluence would be extremely small. Moreover, at such redshifts star formation is not sufficiently progressed and rare stars are too young to explode. On the other hand, up to redshift ~ 10 or so Universe is not cleared and optical depth is short, thereby significant attenuation of X-ray and Gamma-ray through Compton scattering is expected. Even with a more sensitive instrument it would be unlikely to observe a GRB at z >~ 12-15. Therefore, authors should modify their estimation of observable redshifts to more realistic values.
Reviewer 4 Report
English must be thoroughly revised, I suggest a list of corrections that is far from exhaustive.
In the text often there is too much technical detail in hardware description while important information are missing. In particular the results of fig.5 and 6 are not reported in any detail. while Fig.3 seems to bring no useful info.
Results must be reported quantitatively non qualitatively and that is missing: resolution in keV versus the energy, efficiency.
No mention of time resolution is reported, why?
P1 l46: 8x8?? It is not 6x6?
P1 l52: 'responses' -> 'the responses'
P2 l54: 'crystal ...' -> 'crystals to be good candidate as ...'
P2 l73: ' a new data' -> 'new data'
P2 l75: 'recently' -> 'recent'
P2 l 78: 'example ..'-> 'example detection ...'
P2 l79: ' for ...' -> ' for selecting'
P2 l80: 'They ..' Who is they? Future missions?
P2 l86: 'about less than' use the symbol < approximate
P2 l87: 'on trigger' -> 'following a trigger'
P2 l88: 'operated succesfully as well' remove
P2 l90: 'bandwidth' -> energy range'
P2 l101: 'suppliance' -> 'supply'
P3 l115: 'while looking' ->'covering'
P3 115-116: 'its accuracy is 10 armin ...7 sigma'. What does it means exactly, please be clearer.
P3 116-118 please rewrite
P3 l119:' the previous' remove
Table 1: 'Accuracy' specify what is 10 arcmin and sigma. the angular resolution is 10/7 arcmin? If so write it explicitly.
P4 L124-130 and 2.1 (l138-143) should be merged together, ther eare repetitions and contraddictions (pixel size?).
P4 l140: 'for the side that meets the MAPT'- > 'the bottom one'
P4 l150: 'radiation hard'. Why do you need it? How much? Reference to some data.
P4 l154: 'its spectral response range' -> 'its maximum'. Do you have a plot?
P4 l155: pixel size already quoted previously
P5 l166-167: 'The input voltage ...' remove it
P5 162-167: no reason to explain how a PMT works. Remove and/or simplify the text.
P5 l168:200: Add a scheme with analog readout; add a scheme with digital readout. Remove technical details like l178-l181: 'Each analog ...'
P5 l202: 'Using the second set ..' What is it?
P5 l203-204: 'Because ...light, so' Obvious remove it.
P6 l209: 'the X-ray' -> 'a Amtex X-ray'
P6 209-211: what are the energies quoted? The HV applied to the tube or the maximum of the spectra?
Figure 3: are the flux normalized? Why the 15.9 keV is so broad?
Which conclusions do you draw from this picture? Apparently nothing.
P7 l231-232. 'The x and y ...' It is already written on the picture.
P7 l238-289 'of X-ray down to' -> 'to X-ray energy down to'
P7 l240: '200 keV' from where this number comes from?
P7 l240-241: thickness would worsen the resolution due to position dependency of the energy measurement. The conclusion is not obvious and not required by this paper.
P7 Chapter 4: ther is no mention of efficiency versus X-rayenergy.
Fig.5 and 6: 'radioactive' -> 'X-ray'
The fit parameters of the function must be reported on the pictures, especially the resolution.
What are due to the events above 50 ADC in Fig.5 and above 10 in Fig.6
P9 l264-266: 'Therefore ....' Superfluous, already told before.
Round 2
Reviewer 1 Report
Just a minor comment:
- line 197: "Error! Reference source not found" message shows in the draft. Please fix it.
Author Response
We appreciate the referee’s finding of that mistake. We correct it.
Reviewer 2 Report
The authors have made a thorough effort to improve along the lines highlighted in the first referee report.
I do however find some remaining issues that need to be addressed:
(1) The manuscript needs to go through not only another round of thorough spell checking and language editing, but also needs improvements in the clarity and precision of the text and the statements made. Examples:
- Abstract, Lines 57 to 59: "We found that the X-ray detector can measure ... , from which we expect..." --> It should I think rather be broken down into two sentences. One describing the finding, the other one saying what the resulting expectation is.
- Line 64: "even after the dark age" --> do you not rather mean either "immediately after the dark age", or "even opening up parts of the dark age"?
- Lines 66 and 67: I think GRBs have not firmly been associated with UHE particles; they for sure are accelerators, but to what extent and energies remains to be seen.
- Lines 80 and 81: "large volumes of GRBs" --> maybe you mean "deeper statistics" or the likes?
- Line 97: "target them" --> "target it"
Line 102: "with no GRB data" --> missing e.g. "registered"
Line 197: obvious compilation error regarding a reference source
(2) There should be a clear and explicit statement made either up front in the abstract or somewhere between sections 3 and 4 which experiments and measurements were made on which unit of the hardware, so on either the flight model or the engineering model.
(3) Lines 294 to 301: This paragraph is difficult to understand, either really unclear or not entirely correct. Everything else being the same, more distant sources will be redshifted and appear dimmer. Sure, there may be sources that are distant but intrinsically bright, but as the authors state in the introduction, it is rather about detecting a large enough number of bursts. Please clarify this paragraph.
(4) Conclusion: Please also make here clear to the reader on which model of the hardware this study was performed.
Reviewer 4 Report
The english has been substantially improved and now is acceptable.
Figure 4 uses data available from following figures 6 and 7. Those figures should be positioned before Fig.4.
Are there errors on Fig.4? The fit parameters must be reported with errors as well.
Also in Fig.3 the averages must be reported with errors.
Line 197 there is a mistake in referring a reference.
Line 201: FWHM does not need a definition, it is well known.
Line 203: 'tail of the spectrum' -> 'tails of the spectra'
The reason of the tails are unexplained in particular for 13.4 keV.
Line 228: 'value' -> 'values' or simply 'a and b are' (not were)
Line 232-233: when the energy increases up to MeV the interaction mechanisms of photons changes (move to to Compton). It is not clear that increasing the thickness will bring much improvements in efficiency.
The functions use for the fit in Fig.6 and 7 should be written explicitly.
E.g. fig.6 looks like a triple Gaussian.
Figure 4: the independent variable is the energy that must be on the x-axis and the measured variable must go on the y-axis. You must invert the axis.
Line 308: 'from a few keV 307 to MeV' there is no proof that the material is effective in detecting MeV photons. The data are only up to 60 keV.
